# Interface Dynamics and the Influence of Gravity on Droplet Generation in a Y-microchannel

**DOI:** 10.3390/mi13111941

**Published:** 2022-11-10

**Authors:** Alexandra Bran, Nicoleta Octavia Tanase, Corneliu Balan

**Affiliations:** REOROM Laboratory, Faculty of Energetics, University Politehnica of Bucharest, Splaiul Independentei 313, 060042 Bucharest, Romania

**Keywords:** microchannel, immiscible liquids, droplet length, plug flow, Stokes flow, interface, capillarity, gravity

## Abstract

The present experimental investigation is focused on the influence of gravity upon water-droplet formation in a Y-microchannel filled with oil. The flows are in the Stokes regime, with very small capillary numbers and Ohnesorge numbers less than one. The study was performed in a square-cross-section channel, with *a* = 1.0 mm as the characteristic dimension and a flow rate ratio κ in a range between 0.55 and 1.8. The interface dynamics in the vicinity of breakup and the transitory plug flow regime after the detachment of the droplet were analysed. The dependence of droplet length *L* was correlated with the channel position against the gravity and κ parameters. The results of the work prove that, for κ=1, the droplet length *L* is independent of channel orientation.

## 1. Introduction

Investigations of complex fluid phenomena have increased since 1960 [1] thanks to miniaturization technologies, leading to a brand-new area called microfluidics [2,3]. Using small amounts of substances, reducing experimental waiting times, the portability and the possibility of embedded systems has increased the interest in this branch of research and led to the development of a whole new series of fabrication techniques [4]. Many applications of microfluidics are related to the generation and control of droplets between immiscible fluids. In particular, droplet-based microfluidics is widely used in several fields, such as medicine, chemistry and biology. These applications involve mixing fluids in different micrometric junctions, T-shaped [5,6,7,8], cross-shaped [9,10,11] and Y-shaped [12,13] geometries being some of those most commonly used. In this study, the two liquids considered were focused in a Y-shaped 45° bifurcation. The angle between the two inlets is an important parameter for the stream, along with Reynolds and capillary numbers, channel symmetry and cross section [14]. 

As mentioned above, droplet formation in microchannels is used widely in the medical field. Digital polymerase chain reaction (dPCR) was enhanced by using droplets of oil in aqueous phase in order to separate, analyse and quantify nucleic acids [15] in a process called droplet digital polymerase chain reaction (ddPCR). Via this method, the rapid diagnosis of tuberculosis by detection of *Mycobacterium tuberculosis* using exosomal DNA and the diagnosis of occult HBV infection by identifying the covalently closed circular DNA from hepatocytes are possible [16,17].

In addition, droplet-based microfluidics provides a background for single-cell analysis through the encapsulation of individual cells in droplets. This enables the study of the interactions between cells, useful in personalized cancer treatment [18,19], and between cells and different drugs [20]. Droplets can be used as reaction chambers for biomolecule concentration monitoring [21,22,23], protein analysis [24,25] and as mini-incubators for cell culture [26].

In all droplet-based applications, the most important factor is the control of flow regime, in particular the dynamics of the transition from jetting to dripping, the thread breakup/pinch-off location, the frequency detachment of droplets and their dimensions [27,28,29]. Usually, for a given micro-configuration, studies focus on the influences of the material properties of two fluids in contact and the ratio between their input flow rates on the phenomenon in question. Investigations of gravity’s influence on droplet detachment at micro-scales are scarce in the literature, the effects of mass forces on flows being mainly studied in relation to interface stability [30,31]. In [32], the influence of gravity on thread pinch-off, droplet detachment and plug flow onset were investigated at a mesoscale, with a capillary diameter of 2.38 mm.

However, gravity might have, even at micro-/mesoscales, a relevant influence on the stability of a flow regime. In particular, the orientation of the microchannel can affect the formation and dynamics of droplets. The goal of the present work was to investigate the influence of gravity on the plug flow generation in a Y-channel with a square cross section with *a* = 1.0 mm. The channel characteristic dimension was at the upper limit of the micro-scale, at the transition to the mesoscale. The samples investigated were water (dispersed phase) and oil (primary/continuous phase). The two liquids are immiscible, with relatively similar densities and a viscosity ratio of 55 between continuous and dispersed phases. The main parameter in our study was the dependence of the droplet length in the plug flow regime on the flow rate ratio and the channel orientation against gravity. 

## 2. Materials and Methods

### 2.1. Microgeometry—Design and Technological Aspects

The microchannel was fabricated using the soft lithography and double polydimethylsiloxane (PDMS) casting methods. The positive mould was obtained using laser-cutting technology and a 1.0 mm stainless steel sheet. Into this mould, PDMS (Sylgard^®^ 184 from Dow Corning Corporation, Midland, MI, USA) with 10:1 base and curing agent were respectively poured. This elastomer was chosen considering its transparency and spatial resolution. The bubbles formed in the mixing process were removed using a vacuum desiccator obtained from Kartell LABWARE (Milan, Italy). After pouring the PDMS, the polymerization and thus the solidification took place in an oven at 80 °C for 15 min. After cooling down, the PDMS negative mould was carefully peeled off and rinsed with ethylic alcohol 70% (*v*/*v*) for dust removal. There are several methods reported in the literature for double PDMS casting to avoid self-adhesion between the master mould and the second PDMS layer, such as oxygen–plasma treatment followed by methanol passivation [33], self-assembled monolayer formation [34] or hydroxypropylmethylcellulose treatment [35]. We preferred a much simpler method, namely, the coating of the PDMS mould with a thin layer of immersion oil to facilitate the PDMS-PDMS casting. Newly prepared PDMS was poured into the mould and kept at room temperature for 48 h. We avoided applying a high temperature at this stage because it could blend the two polymer layers, making it impossible to set them apart properly. After the second casting, the PDMS channel was immersed in ethylic alcohol 70% (*v*/*v*) for 10 min for oil dissolution and dried with compressed air. The inlet and outlet holes were subsequently made using a puncher. 

The microchannel, along with a glass microscope slide, had undergone a plasma treatment (Harrick Plasma Cleaner PDC-002-CE, Ithaca, NY, USA) for 2.5 min that allowed adhesion. It has been stated that a longer plasma treatment changes the wettability of the PDMS surface [36]. In Figure 1, we show the importance of time in our experiments. A longer time means hydrophilic surface behaviour and results in visible changes in the interface between the liquid and the solid channel walls. We considered 2.5 min to be an adequate time for the drop-shaped interface and sufficient bonding. Afterwards, metallic ports were attached using a two-component epoxy adhesive. 

By means of this technique, a microchannel with a square cross section, with 1.1 mm sides (the two inlets and the outlet) and a 45°-angle symmetrical Y-bifurcation, was created, as shown in Figure 2. The aspect ratio of the microchannel is crucial for the flow regime. For example, at low Reynolds numbers for the oil phase (Re << 1), low aspect ratios of 0.125 lead to jetting regimes, with no interface instabilities for plug formation [37].

### 2.2. Fluid Samples

The previously described microchannel was filled with two immiscible Newtonian fluids, deionized water and sunflower oil, whose properties were determined at 25 °C (see Table 1).

The densities were measured by gravimetric procedure using a 0.5 μl Hamilton syringe and an AS 82/220.R2 analytical balance obtained from Radwag (Radom, Poland). The interfacial tension was determined using the pendant drop method, and the viscosities were measured with the MC301 Anton-Paar rheometer (Graz, Austria), using the con-plate configuration. Wettability tests were conducted using a Drop Shape Analyzer DSA25S obtained from Krüss Scientific (Hamburg, Germany), by the sessile drop method, automatically dosing 3 μl drops. For interface finding, a polynomial fitting was adopted. The results presented in Figure 3 show that both the PDMS and glass displayed hydrophobic behaviour (more relevant for PDMS), but they were both oleophilic, since the oil–substrate contact angle was below 90°. 

### 2.3. Experimental Setup and Visualizations

The setup used in the experimental investigations is shown in Figure 4. The microchannel was fixed on a 3-axis micrometre stage between the diffusive uniform light source and the fast camera, with the possibility of being orientated in 4 positions (Figure 5): (i) oil above–water below; (ii) water above–oil below; (iii) vertical; and (iv) horizontal. The visualizations for the horizontal position of the microchannel were obtained using a mirror with an adjustable inclination (Figure 4b). In all the tested positions, the microchannel was initially filled with oil (primary phase), and the dripping flow regime was recorded in water. 

The volumetric constant flow rates of water $Q_{w}$ and oil $Q_{o}$ were transported in the two branches of the microchannel by a dual Harvard syringe pump. The visualizations were performed with a Phantom VEO 710 fast camera (Vision Research, Wayne, NJ, USA) with up to 5300 fps and a minimum resolution of 1920 × 1080 pixels. The corresponding dimension for 1 pixel was 2.5 μm. The camera was equipped with a microscopic tube connected to an MX-2 objective obtained from Edmund Optics.

The input flow rates in each branch were in the range of Qϵ [500, 900]μL/h, which corresponded to a maximum velocity of 0.2 mm/s (and 0.4 mm/s after bifurcation). There were 25 dripping cases tested with the ratio $k=Q_{w}/Q_{o}$, with k ϵ [0.55, 1.8] at the viscosity ratio $\mu =\eta_{o}/\eta_{w}$, μ=55. Low flow rates, along with dominant interfacial forces, were responsible for slug regimes [38]. A series of images were selected from the video recordings of the region of water-droplet breakup and droplet formation in the main channel (Figure 6a–f); a detail of the interface dynamics associated with droplet breakup is shown in Figure 7. In Figure 6, the water was seeded with 10-micron Ag particles to visualize the internal vortices in the droplet generated by the advance of the interface and finally its rupture. The present analysis is focused on the dependence of the elongated droplet length *L* on the flow rate ratio κ and the channel position (Figure 5, Figure 6 and Figure 7) and the local kinematics in the vicinity of the breakup, represented in Figure 6e and Figure 7: (i) neck velocity $v_{n}:=dw_{n}/dt$ (where $w_{n}$ is the minimum thickness of the interface); (ii) the wall slip velocity of the droplet $v_{A}$ after the rupture of the interface.

## 3. Results and Analysis

### 3.1. The Influence of Gravity on Droplet Length

In the range of the tested flow rates, the characteristic non-dimensional numbers are listed in Table 2. As expected, the droplet dripping was dominated by surface tension, the Weber and capillary numbers being very small, the phenomena taking place in the Stokes flow regime (the so-called slug regime [38]). The viscosity ratio of the two fluids was relatively large in our case; at the tested values of the capillary numbers (Table 2), the droplet mobility was close to unity, which indicates that the droplet filled the channel almost completely and that the thickness of the oil layer was below the micron level [39]. Therefore, the width of the droplet was approximated by the *a* dimension.

The range of values for the capillary number indicates the dripping regime with elongated droplets, i.e., $\bar{L}=L/a>1$ [40,41]. We have to remark that the contribution of friction between water and oil does not have a negligible influence, since Oh>0.1. The Bond number is below 0.1—a relatively large value for the water–oil flow in a microchannel due to the imposed dimension of the space scale (defined by the value of a).

The measured dependencies $\bar{L}\left(\kappa \right)$ as functions of the channel positions shown in Figure 5 are displayed in Figure 8. The dispersion of the experimental data at κ=constant was relatively high, variations in droplet length being recorded in some cases. Probably, this phenomenon was generated by the inertial influence: increasing the velocity, i.e., the capillary number, means decreasing the droplet length [34]. Such variations were observed at constant κ and some flow-rate values, even if the capillary number was kept low (Table 2). The dispersion of the measured droplet length might also have been determined during our experiments by the presence of pulsed/unsteady flows, which were mainly recorded at low-flow-rate regimes. Consequently, the R-squared coefficient from the linear regression was low,$\;R^{2}\in \left(0.33\;÷0.65\right)$, except for the vertical case, where the linear fitting was close to horizontal and $R^{2}=0.015$. However, the linear fitting clearly indicated two important results of our experimental study:

Gravity has a relevant influence on $\bar{L}\left(\kappa \right)$ dependence, the length$\;\bar{L}$ increasing with κ; the slope $\bar{L}$ against κ was almost zero for the vertical position, the highest value being observed in case (ii), shown in Figure 8, i.e., a less dense liquid at the bottom;At κ=1, the influence of gravity was absent; the orientation of the channel did not influence $\bar{L}.$

### 3.2. Interface Breakup Dynamics and Droplet Formation

The rupture of the water interface and the dynamics of droplet formation in the vicinity of breakup were investigated at $\kappa =1,\;Q_{w}=Q_{o}=900\;\mu L/h.$ The variations in the minimum interface thickness$\;w_{n}$ and the neck velocity $v_{n}$ over time showed similar patterns to those observed in some previous studies of the Newtonian breakup filaments of droplets [42] (Figure 9): an exponential increase in $v_{n}$ previous to interface rupture and droplet detachment.

Immediately after the interface rupture and the droplet detachment, the water–oil boundary was recorded at 5200 fps, the image processing being performed in ImageJ and MATLAB software (MathWorks, Natick, MA, USA) [42] and displayed in Figure 10. The steady state, defined as the kinematic condition $x_{A}=x_{B}$ (Figure 7), was achieved after approximately 110 ms from the breakup.

Using the data from Figure 10, the time variations in (i) space position $x_{A}$, (ii) slip velocity $v_{s}$ and (iii) contact angle $\alpha^{o}$ are represented in Figure 11.

In Figure 11, three flow regimes related to the post-breakup dynamics of point A are shown:R1. Sharp decrease in the slip velocity and increase in the contact angle;R2. Constant slip velocity and contact angle (domain where the slip velocity is proportional to the product between the capillary velocity and $\alpha^{3}$) [43];R3. Slide increase in contact angle and small oscillations in slip velocity. At the end of the R3 regime, the droplet interface reaches the steady state, i.e., the plug flow.

## 4. Conclusions

The experimental study investigated the influence of gravity on dispersed-phase (water) droplet length in oil (continuous phase) during a plug flow regime in a square-cross-section Y-microchannel with a characteristic dimension of a=1.0 mm. The imposed flow rates were in the range of [500 ÷ 900] μL/h, and the viscosity ratio between the fluids was μ=55. The Ohnesorge characteristic number was below one, *Oh* ≅0.34, and the capillary number for all experiments was below $5\times 10^{-3}$.

Taking into consideration different channel positions against gravity, increasing droplet lengths *L* were generally recorded while increasing the flow rate ratio κ, κ∈[0.55÷1.8] (Figure 8). The influence of gravity on the dependence of droplet length on the flow rate ratio was remarkable, especially for the less dense liquid (oil) below water orientation (ii), shown in Figure 5 and Figure 8. In this case, the buoyancy force was a destabilization force against friction; the ratio between these two forces defines the Rayleigh number [43]. This number might be the non-dimensional parameter related to the influence of microchannel orientation on interface dynamics. Almost no effect of gravity was observed in the case of the vertical position of the channel, where gravitational acceleration had the same orientation as the flow velocity. 

One main conclusion of the experimental investigations is that channel orientation against gravity has no influence on *L* at κ=1 (the *L* dimension is dependent on the capillary number, but this aspect was not the focus of our study). This result does not have a theoretical or numerical justification at present. However, it is an important result for applications where the effects of gravitational force may be in question and the control of microchannel orientation is difficult. The plug flow regime remains the same, with unchanged droplet length, independently of channel orientation, if the flow rate ratio of the fluids is maintained in the vicinity of one.

The correlation of the experimental data reported here with 3D numerical simulations is under investigation in our laboratory [37,42]; see also [44]. The laminar flows (with gravitational terms included) are being solved using the VoF code implemented in the ANSYS software package (Ansys, Inc., Canonsburg, PA, USA). The first results are promising but not conclusive; the computation time for each case takes at least 2–3 weeks on our machines (since the time step for convergence is 0.05 ms). Therefore, we decided to include in the present paper only the experimental results; a separate paper will be dedicated to the corresponding numerical solutions.

The analysis of the post-breakup dynamics of the interface and droplet formation revealed three flow regimes until the plug/steady flow was reached. The investigation of these transitory regimes (Figure 10 and Figure 11), in relation to neck velocity (Figure 9), and the material properties of the samples is the goal of our next work.

## Figures and Tables

**Figure 1 micromachines-13-01941-f001:**
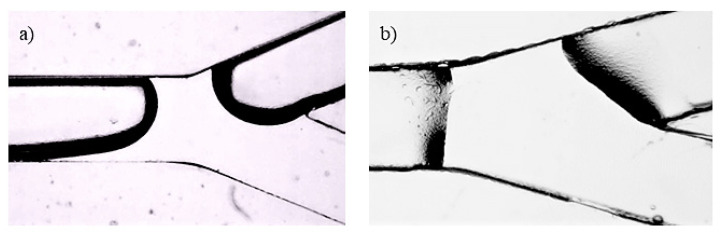
Influence on the water–wall contact of the polydimethylsiloxane (PDMS) time duration treatment in plasma: (**a**) 2.5 min; (**b**) 5 min.

**Figure 2 micromachines-13-01941-f002:**
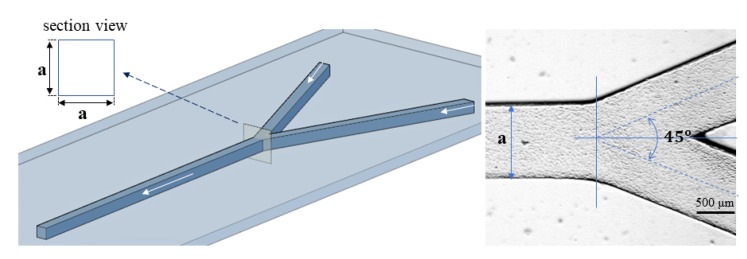
Geometry of the tested microchannel; microscopic detail of the Y-bifurcation with a ≅1.0 mm.

**Figure 3 micromachines-13-01941-f003:**
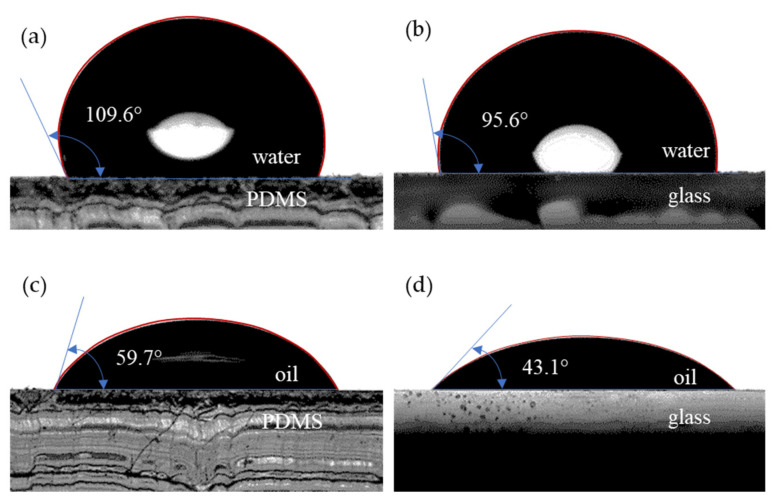
Contact angle: (**a**) water–PDMS; (**b**) water–glass; (**c**) oil–PDMS; (**d**) oil–glass.

**Figure 4 micromachines-13-01941-f004:**
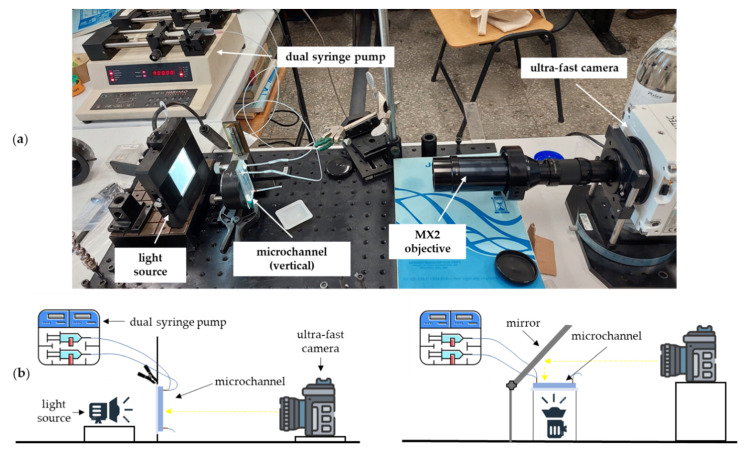
Experimental setup: (**a**) general view; (**b**) positions of the microchannel.

**Figure 5 micromachines-13-01941-f005:**
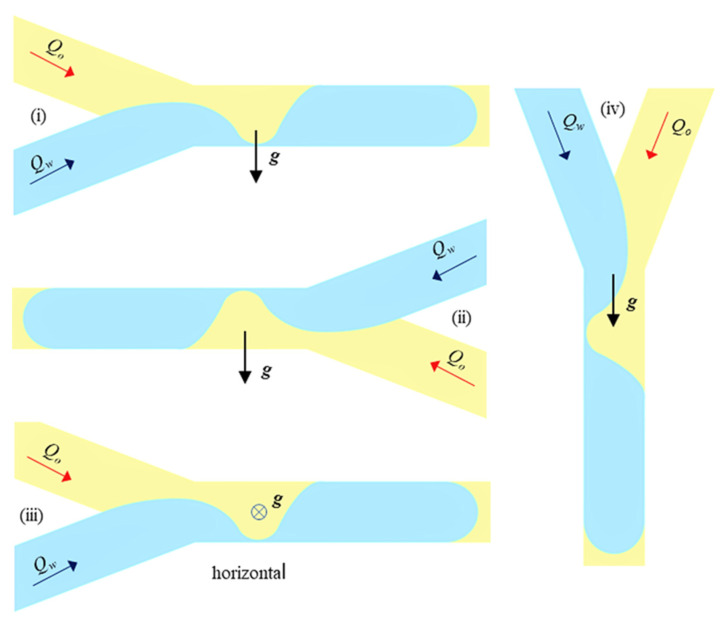
The four tested orientations of the microchannel: (**i**) dense liquid below; (**ii**) dense liquid above; (**iii**) horizontal; (**iv**) vertical ($Q_{w}$= water flow rate, $Q_{o}$ = oil flow rate, g = gravitational acceleration)

**Figure 6 micromachines-13-01941-f006:**
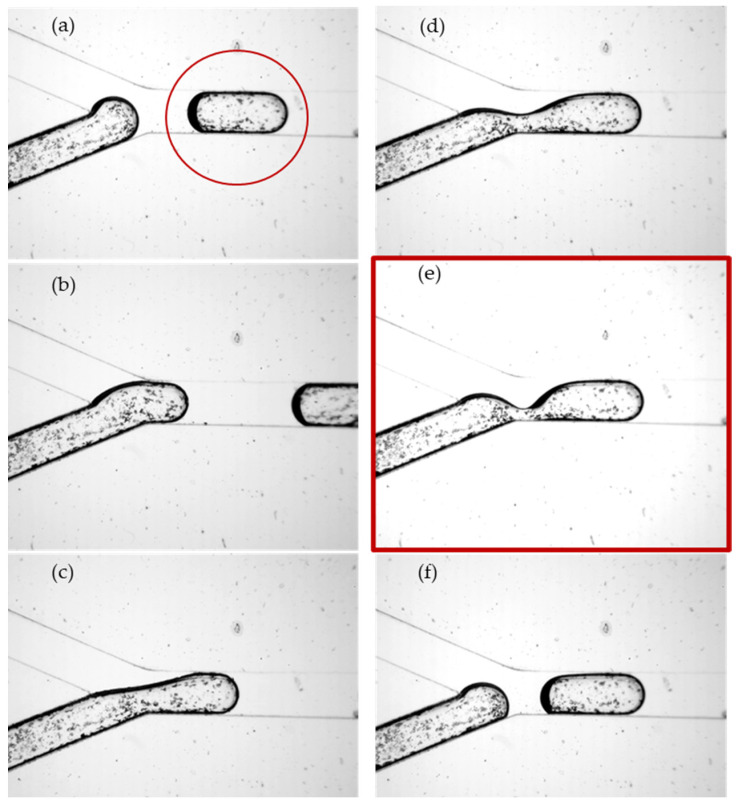
Interface dynamics and water-droplet formation in oil for one cycle; series of images captured from video recordings at 1000 fps: from droplet detachment (**a**,**f**) to the interface rupture (**e**). (**a**) previous droplet marked in red circle; (**b**) water advancement; (**c**) neck formation; (**d**) neck thinning; (**e**) neck before interface rupture; (**f**) droplet after interface rupture.

**Figure 7 micromachines-13-01941-f007:**
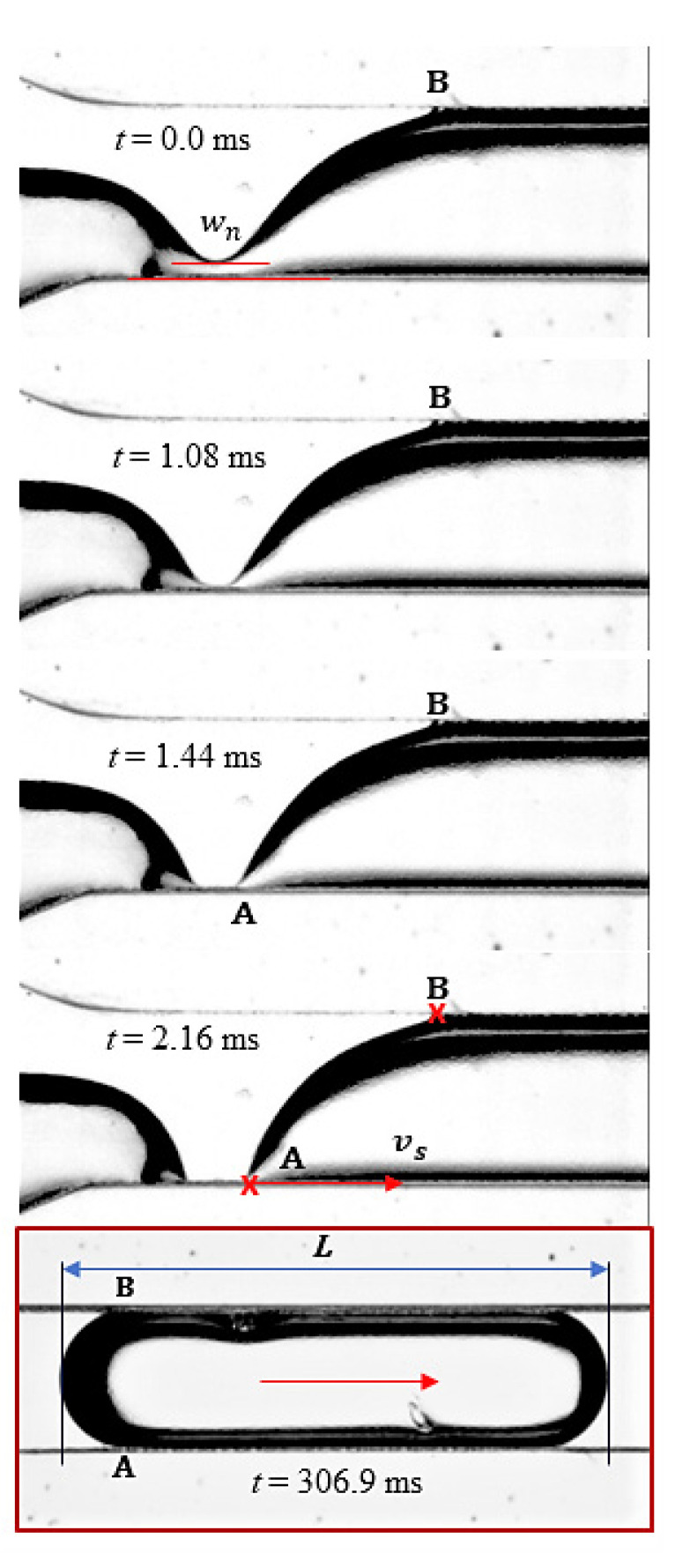
Onset of the dripping mode in the microchannel with elongated droplets (*L/a* > 1); time evolution of the breakup phenomena and the formation of the water droplet; images extracted from video recordings (see Appendix A, at the moments marked in the pictures). Here, the space points A and B are limiting the droplet after the rupture of the interface. The length *L* was measured by image processing of selected pictures using a procedure implemented in ImageJ software. Red arrows indicate point A and droplet velocity direction. Final droplet is marked in the red square.

**Figure 8 micromachines-13-01941-f008:**
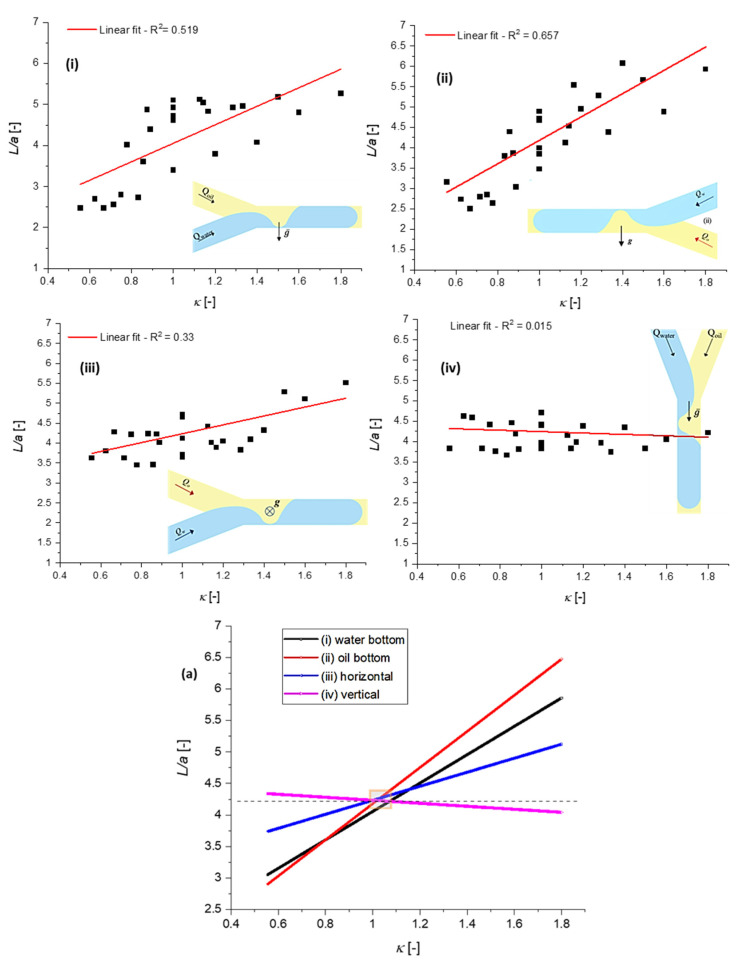
Nondimensional water-droplet length $\bar{L}=L/a$ as a function of microchannel orientation (**i**–**iv**); the dependence $\bar{L}\left(\kappa \right)$ for the tested configurations (**a**).

**Figure 9 micromachines-13-01941-f009:**
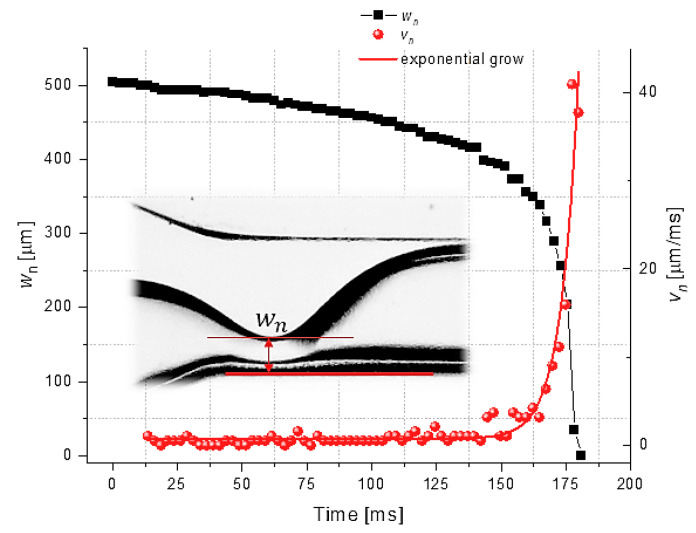
Variation over time in the thickness interface and neck velocity.

**Figure 10 micromachines-13-01941-f010:**
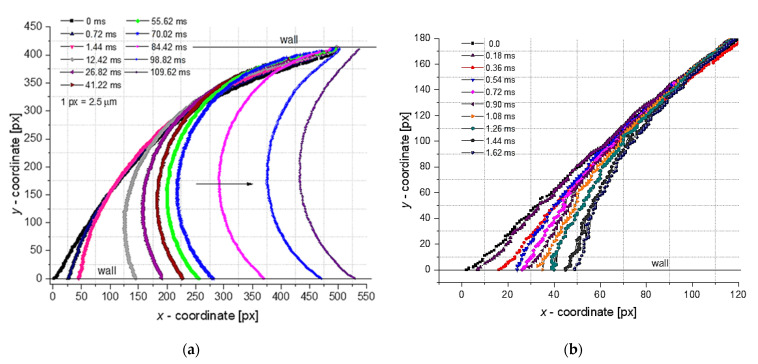
Dynamics of the droplet interface after breakup at 0<t<110 ms, with detail of the interval 0<t<1.7 ms. It is assumed that for time t>110 ms the local geometry of the droplet interface will remain constant. (**a**) interface aspect from interface rupture (0 ms) to droplet symmetry (109.62 ms); (**b**) detail of the interface in the first 1.62 ms after rupture.

**Figure 11 micromachines-13-01941-f011:**
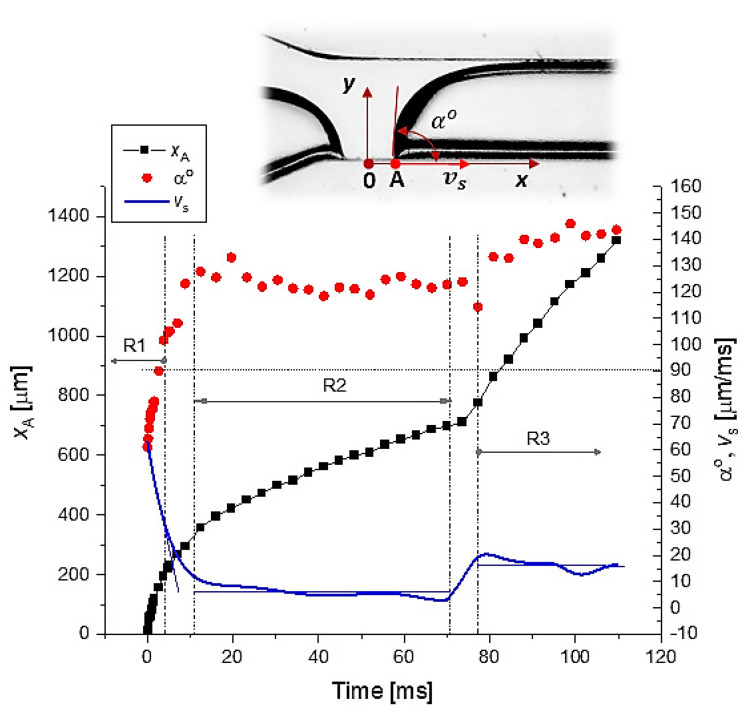
Kinematics after the breakup in the vicinity of space point A; see also Figure 7.

**Table 1 micromachines-13-01941-t001:** Properties of the fluid samples.

Fluid	$\mathrm{Density}\;\rho \;(\mathrm{kg}\text{/}m^{3})$	Viscosityη (mPa·s)	Interfacial Tension σ (N/m)
Deionized water Sunflower oil	1000 925	1 55	0.025

**Table 2 micromachines-13-01941-t002:** Nondimensional numbers computed with the material properties of the primary phase (oil); here, $v_{c}=Q_{oil}/a^{2}$.

Symbol/ Name	Relation	Values	Observations
*We*—Weber	$\frac{\rho_{oil}v_{c}^{2}a}{\sigma }$	$<2\times 10^{-6}$	Surface tension dominant against inertia
*Oh*—Ohnesorge	$\frac{\eta_{oil}}{\sqrt{\rho_{oil}\sigma a}}$	0.345	Friction in relative equilibrium with interfacial forces, 0.1 < Oh < 1
*Bo*—Bond	$\frac{\Delta \rho ga^{2}}{\sigma }$	0.035	Surface tension dominant against gravity
*Ca*—Capillary	$\frac{\eta_{oil}v_{c}}{\sigma }$	$<4.4\times 10^{-3}$	Dripping regime with elongated droplets Ca < 0.005
*Re*—Reynolds	$\frac{\rho_{oil}v_{c}a}{\eta_{oil}}$	$<4\times 10^{-3}$	Stokes flow regime

## Data Availability

Not applicable

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
