# Peer review of "Interface Dynamics and the Influence of Gravity on Droplet Generation in a Y-microchannel"

_micromachines, 2022, doi:10.3390/mi13111941_

Round 1
Reviewer 1 Report
Most people use microfluidics on an inverted microscope where gravity effects are negligible, If you want to study gravity, you should rotate the device from a vertical configuration by gradual angles with a rotating stage. You should develop some theoretical ideas on how gravity affects the device or use comsol simulations. If you want to study gravity, there should probably be less attention to other aspects. How do the dimensionless constants come into play with device orientation? Is there any practical application of gravity on your microfluidic device?
Author Response
Dear reviewer,
Thank you for your comments and suggestions, as they are useful for improving the present paper and the focus of our further investigations.
The experimental setup consisted of a camera equipped with a microscopic tube connected to a MX-2 objective and the microchannel was placed on a 3-axis micrometre stage that allowed the orientation on right angles. The microchannel was kept perpendicular to the camera, with exception of the horizontal position where a mirror was used, see Fig. 4.b. We haven’t used a microscope. Your suggestion to take into consideration a rotational stage and to report the influence of different angles orientation of the microchannel on the dripping regime is definitely of interest, but is not at the moment an aim of the work.
Our study is exclusively an experimental one. Recently we started the 3D-simulations for the investigated configurations using VoF code implemented in ANSYS, the gravity force being included in the Navier-Stokes equation for each orientation. The first results are promising but not conclusive; the computation time for each case will take at least 2-3 weeks on our machines (since the time step for convergence is 0.05 ms). Therefore, we decided to include in the present paper only the experimental results and a separate paper will be dedicated to the corresponding numerical solutions.
In the case of orientation from Fig. 8.ii (the less dens liquid below), the buoyancy force is a destabilization force against friction (at Bo = 0.035); the ratio between those 2 forces defines the Rayleigh number. This number might be the non-dimensional parameter related to the influence of microchannel orientation on the interface dynamics. It will be also of interest to investigate the influence of buoyancy force on the stability of the dripping regime and droplet length, respectively. We hope to treat this subject in a chapter of the future paper dedicated to numerical simulations.
The result of the present study is that flow regime in such microchannels is independent of the orientation if the input flow rates ratio is around one. This observation is useful for applications where channel orientation cannot be fully controlled, and the droplet length is of considerable interest to be kept in a desired interval.
Reviewer 2 Report
The manuscript can be accepted after a few modifications. Some minor corrections, which are listed as follows, should get completed.
1. How does the length of the droplet L of the iv position change with k? How dependent is gravity on it at this time?
2. The conclusion mentions that the direction of the microchannel does not affect the length of the droplet L. How to prove it?
Author Response
Dear reviewer,
Thank you for the observations regarding the present work. We analysed your comments and find them useful for making more clear the presentation of our paper.
- The flow rate ratio has almost no influence on L for channel orientation (iv) in Fig. 5, since gravity acts along the flow direction. This expected result is observed in Fig. 8.a. However, we enclose a new figure in Fig. 8 for the experimental data of this orientation.
- The study was conducted experimentally, so we did not cover any theoretical approach of this topic. For the range of tested flow rates, the only case where the channel orientation has to influence on droplet L is the vertical case, as can be observed in Fig. 8.a. This result is an experimental finding, at the moment we do not have a theoretical justification of it. First, we are looking for the correlation to the numerical solutions (work in progress).
Some parts of our answers are introduced in the reviewed version of the paper, mainly in the Conclusion chapter (the changes are marked).
The English language was checked, and some minor spelling mistakes were removed.